# Phenotypic and Genetic Variation in Morphophysiological Traits in Huanglongbing-Affected Mandarin Hybrid Populations

**DOI:** 10.3390/plants12010042

**Published:** 2022-12-22

**Authors:** Qibin Yu, Fanwei Dai, Riccardo Russo, Anirban Guha, Myrtho Pierre, Xiaokang Zhuo, Yuanzhi Zimmy Wang, Christopher Vincent, Frederick G. Gmitter

**Affiliations:** 1Citrus Research and Education Center, Institute of Food and Agricultural Sciences, University of Florida, Lake Alfred, FL 33850, USA; 2Institute of Sericulture and Agricultural Products Processing, Guangdong Academy of Agricultural Sciences, Guangzhou 510610, China

**Keywords:** *Candidatus* Liberibacter asiaticus, citrus breeding, *Citrus reticulata*, photochemistry, selection for tolerance

## Abstract

Huanglongbing (HLB) caused by ‘*Candidatus* Liberibacter asiaticus’ (*C*Las) is the most costly disease for the global citrus industry. Currently, no effective tools have been found to control HLB. Most commercial citrus varieties are susceptible to HLB, though some citrus hybrid cultivars have reduced sensitivity to the disease. Citrus breeding populations contain a large diversity of germplasm, with thousands of unique genotypes exhibiting a broad range of phenotypes. Understanding phenotypic variation and genetic inheritance in HLB-affected mandarin hybrid populations are crucial for breeding tolerant citrus varieties. In this study, we assessed 448 diverse mandarin hybrids coming from 30 crosses, and 45 additional accessions. For HLB tolerance, we measured HLB severity visual score and *C*Las titers by qPCR. We also measured seven morphophysiological traits indirectly related to HLB tolerance with leaf area index (LAI), leaf area (LA), leaf mass per area (LMA), photosystem II parameters (Fv/Fo, Fv/Fm), and photochemical performance index (PI_abs_). By estimating the genetic variation in five half-sib families, we estimated the heritability of phenotypic traits and found a significant genetic effect on HLB visual score and photosynthesis parameters, which indicates opportunities for the genetic improvement of HLB tolerance. In addition, although it is easy to identify infected trees based on HLB symptomatic leaves, visually phenotyping whole trees can be difficult and inconsistent due to the interpersonal subjectivity of characterization. We investigated their relationships and found that LAI was highly correlated with HLB score, with correlation coefficients of r = 0.70 and r = 0.77 for the whole population and five half-sib families, respectively. Photochemical parameters showed significant correlation with HLB severity and responded differentially with the side of the canopy. Our study suggests that LAI and photochemical parameters could be used as a rapid and cost-effective method to evaluate HLB tolerance and inheritance in citrus breeding programs.

## 1. Introduction

Huanglongbing (HLB), associated with the putative pathogen *Candidatus* Liberibacter asiaticus (*C*Las), is the most costly disease for the global citrus industry. Currently, no effective tools have been found to control the spread of the *Diaphorina citri* vector and *C*Las pathogen [1]. Though promising approaches have been reported to manage and combat the disease [2], no definitive solution is at hand. Commercial citrus varieties such as mandarins, grapefruits, and sweet oranges are introgression hybrids of *C. reticulata* and *C. maxima* with a long history of evolution and domestication [3] and all are susceptible to the disease. These commercial citrus varieties are closely related with a narrow genetic base and lack HLB resistance, though specific varieties within this group exhibit greater disease tolerance than others [4].

Citrus breeding populations contain a large diversity of germplasm and thousands of unique genotypes with a broad range of phenotypes. Some citrus hybrid cultivars have been reported to exhibit reduced sensitivity to HLB [4,5]. A desirable long-term solution to HLB would be the industry-wide deployment of tolerant cultivars developed through genetic breeding methods. As HLB has become endemic in Florida, there have been substantial differences in the speed of infection and severity of symptoms in breeding populations [5]. Understanding phenotypic variation and genetic inheritance in HLB-affected mandarin hybrid populations are crucial for breeding tolerant citrus varieties. However, there is no information on phenotype variation and genetic inheritance in affected mandarin hybrid populations. In addition to the lack of knowledge of quantifying genetic variation and inheritance of HLB tolerance, the evaluation of HLB-affected trees in the field has proven to be difficult and inconsistent due to person-to-person implementation of subjective severity scoring techniques [6,7,8]. The complex nature of HLB and genetically diverse citrus germplasm results in large but inconsistent phenotypic differences such as sparse foliation, yellowing foliage, and branch dieback, associated with genotype by environment interactions over years in field trials. Even for a single individual, the grading standards could be affected by the overall situation of the field trials at different times of the year. Therefore, citrus breeders need to find a reliable tool to identify tolerant selections to compare results generated from different studies and years without subjective human errors. Methods of evaluation of tolerance by instruments measuring relevant quantitative traits can reduce cognitive bias.

Previous studies have used instruments to measure physiological and morphological traits for the assessment of HLB, but with a limited number of citrus cultivars either under greenhouse conditions or in commercial citrus groves [9,10,11]. The characterization of phenotypic traits is often time-consuming. When dealing with many diverse genotypes of the citrus breeding population in field trials, only rapid methods are practically applicable. It is essential to test the efficiency of these methods in screening diverse genotypes in the field. A quick, cost-effective method to evaluate HLB tolerance from many citrus accessions in field trials is considered a critical step in developing HLB-tolerant commercial cultivars. 

In this study, we assessed eight morphological and physiological traits based on our results and previous studies [9,10,11,12] and investigated their relationships with a nine-point visual HLB severity grading score in a diverse mandarin hybrid breeding population consisting of 448 diverse individuals with tree ages from 15 to 25 years old. All selected phenotyping traits were considered biologically relevant, relatively low cost, easy and quick measurable traits. We want to know whether data generated from instruments could replace our human visual scoring system, to reduce the human cognitive bias. The goal of the study was to, (1) understand phenotypical variation and genetic inheritance of HLB tolerance, (2) identify relationship of morphophysiological traits with HLB visual score, (3) establish efficient and effective tools for assessment of HLB tolerance in large and diverse citrus breeding populations in field trials.

## 2. Materials and Methods

### 2.1. Plant Materials

Field-grown trees of mandarin hybrids (*Citrus reticulata*) with various genetic backgrounds were measured at the University of Florida Citrus Research and Education Center (UF-CREC). In this study, we use the ‘mandarin’ term to refer to commercial or consumer’s designation as small, easily peeled sweet fruit, although mandarins or mandarin hybrids have different proportions of admixture from ancestral pummelo species (*Citrus maxima*) [3]. The trees were grown under the same environmental conditions of soil, irrigation, and illumination. Measurements were carried out in October of 2019. In total, 448 trees were investigated (Table 1) from 30 crosses and 45 additional individual selections that were between 15 to 25 years old (Table 2). All mandarin hybrids have various genetic backgrounds due to years of cross and breeding efforts among mandarins, oranges, and grapefruits. 

### 2.2. HLB Visual Grading

Each tree was rated for degree of HLB severity using a 1-to-9-point scale, where 1 was for the poorest trees and 9 was for only the ideal or an exceptional tree (Figure 1). The 9-point scale method was modified from our previous 6-point visual grading scale [13]. This rating is a compilation of all aspects of the citrus tree health relating to HLB including overall tree green color, canopy density, and branch dieback. The final visual grading is for overall health due to combinations of different factors. One person was assigned to assess all trees in the field.

### 2.3. PCR Detection of CLas

Total DNA was extracted from leaf midribs and petioles using the Plant DNeasyMini Kit (Qiagen, Valencia, CA, USA) according to the manufacturer’s instructions. During the period of HLB disease evaluation in October 2019, at least seven fully expanded mature leaves were randomly collected from different branches and different quadrants of each tree. Real-time qPCR (RT-qPCR) assays were performed as previously described [14]. The RT-qPCR was performed on an ABI 7500 thermocycler with probes specific to *C*Las 16S ribosomal gene and citrus cytochrome oxidase gene. The data were expressed as the DNA amplification cycle threshold (*Ct*). The *Ct* value threshold for *C*Las-infection <33.

### 2.4. Measurement of LAI 

Citrus tree canopy cover or vegetative leaf area was measured in terms of leaf area index (LAI) by using AccuPAR LP-80 (Meter Group, Pullman, WA, USA). An external PAR (photosynthetic active radiation) sensor was placed in an open area and another PAR probe (LP-80 instrument) was placed under the canopy of the tree. The LP-80 computed LAI from the readings in the open and below canopy PAR and χ (leaf angle distribution parameter). The χ parameter was kept at default (χ = 1). On average, 6 to 7 measurements were taken in different directions around each tree for a better approximation.

### 2.5. Leaf Physiological Measurements 

We used chlorophyll *a* fluorescence (ChlF) to detect and quantify citrus trees’ response to HLB. The ChlF signals are highly sensitive to biotic and abiotic stress conditions and give information on the efficiency of leaf photochemistry and photosynthetic system [15]. This technique often provides early detection of stress before the appearance of any visible symptoms. We measured the polyphasic rise of ChlF transient (OJIP) using a portable chlorophyll fluorimeter instrument (OS-30p+, Opti-Sciences, Hudson, NH, USA) that provided rapidity of data acquisition and large-scale assessment of citrus tree leaf physiological conditions and tree health In October 2019, between 9 and 10 am, we randomly harvested 12 mature leaves per tree from each of the four quadrants (East, West, North, and South) of the canopy. Sampled leaves were immediately placed in sealed plastic bags with moistened tissue and transported to the lab in a cooler within an hour for ChlF measurements [16]. Leaves were first dark-acclimated for 45 min, then probed twice on two different sides of the leaf lamina to increase the amount of probed leaf area. A weak modulated red light was used to measure the minimal fluorescence (Fo) and then a saturating red actinic light of 3500 µmol m^−2^ s^−1^ was applied for 3 s to measure maximal fluorescence (Fm). The OJIP test protocol allowed calculations of variable fluorescence (Fv = Fm − Fo), the maximum quantum efficiency of PSII (Fv/Fm), Fv/Fo (variable fluorescence normalized over F_O_), performance index per absorbance (PI_abs_), and various components of PI_abs_ which were calculated according to Stirbet and Govindjee [17]. Details on the calculation and meaning of the selected OJIP test-derived parameters used in this study have been included in Appendix A.

Leaf area (LA, cm^2^), dry weight (DW, mg) and leaf mass per area (LMA = DW/LA) were measured and calculated on the same leaves used for ChlF measurements. LA was measured individually per leaf (whole leaf) using a leaf area meter (LI-3100, LiCor, Inc., Lincoln, NE, USA). DW was also measured individually after drying in an oven at 60 °C for 72 h followed by weighing using an analytical balance [12].

### 2.6. Statistical Analysis

Morphological and physiological data were analyzed using JMP Pro 15 (SAS Corporation, Cary, NC, USA). Mixed model was used to obtain the variance components of family $\sigma_{f}^{2}$ and $\sigma_{e}^{2}$. Heritability was estimated as
(1)$$h^{2}=\frac{4\sigma_{f}^{2}}{\sigma_{f}^{2}+\sigma_{e}^{2}}$$

Effects were tested using analysis of variance with command Means/Anova. Tukey–Kramer Honestly Significant Difference was used for multiple means comparisons using α = 0.05. Pearson correlation coefficients of phenotypical traits were calculated. Variation in traits was identified by using principal components analysis (PCA). Simple linear regression was used for predictive modeling to predict the value of HLB visual score by LAI. The ChlF variable PI_abs_ correlated with HLB rating was deemed biologically relevant and was chosen to test for association with HLB score using a linear model which included fixed effects for HLB rating, canopy quadrant, and their interactions. These tests were performed using the lm() command in base R. Models included quadrant and HLB rating as fixed factors and the sample was nested. Different yields that contributed to PI_abs_ were first assessed using correlation analysis and tested for significance of correlation using Pearson’s test for significance. This was performed using the ‘rcorr’ function of the package (https://cran.r-project.org/web/packages/Hmisc/index.html, access on 1 January 2022) in the R statistical language (https://www.R-project.org, access on 1 January 2022). The yields were then tested using linear regression with the same model structure as that used to test PI_abs_. 

## 3. Results

### 3.1. Genetic Effect on HLB-Tolerant Traits

The number of progenies from 30 crosses varied from 100 individuals for LB 8-9 × Seedless Kishu to 2 individuals in 7 different crosses (Table 1). Five half-sib families with Seedless Kishu as a common pollen parent had different female parents: LB8-9, Daisy, Lee, Temple and LB 7-11. To identify if HLB tolerance or sensitivity is heritable, we compared these five half-sib families. LB 8-9 parent is a well-known HLB-tolerant tree with an HLB visual score of 9, whereas Kishu and Daisy were sensitive to HLB with an HLB visual score of 6. Temple, Lee and LB 7-11 had HLB visual scores of 8, 7 and 7, respectively (Table 2). HLB score, LAI, PI_abs_, Fv/Fm, Fv/Fo and LMA were the highest in the progeny of LB 8-9 × Seedless Kishu than other families (Figure 2). Heritability of Fv/Fm and Fv/Fo were higher than other traits, whereas heritability of LAI was the lowest (Figure 3).

### 3.2. HLB Score and Ct Value

The mean HLB score from the overall diversified population of 448 individuals was 5.87 and the 95% confidence interval was between 5.72 and 6.02. The HLB scores for 45 individual accessions ranged from 3 to 9 (Table 2). LB8-9, LB8-8, and a hybrid of LB 8-9 × Orlando had the highest HLB scores (9), whereas Fortune × Ortanique, LB 5-19, and Shiranuhi had the lowest HLB score (Table 2). For the whole population, only 14 individuals had an HLB score of 9, which accounted for 3% of the population, and 8 of these accessions had either tolerant LB8-8 or LB8-9 as parents. LB 8-8 is a sibling of LB 8-9 and had an HLB visual score of 9. There were 74% positive trees in the first diagnosis in December of 2019. In the second diagnosis in April of 2021, only 47% were positive trees (Appendix A). The Ct values ranged from 20.1 to 40 and 23.4 to 40.0 for the first and second diagnoses, respectively (Appendix A). 

### 3.3. Leaf Morphophysiological Traits

Overall, the collective population was divided into nine groups based on the HLB visual score. Measured leaf traits were generally within the expected values and distributions for citrus leaves (Table 3). LAI ranged from 1.23 to 4.51 and varied significantly (*p* ≤ 0.05) across the HLB score. The LMA values ranged from 0.012 to 0.014 g·cm^−2^. The ChlF variable PI_abs_ ranged between 2.57 and 3.08, whereas Fv/Fm and Fv/Fo showed very marginal changes, ranging between 0.75 and 0.76 and 3.13 to 3.37, respectively. Besides LAI, leaf area, leaf DW and PI_abs_ varied significantly (*p* ≤ 0.05) across the HLB score (Table 3). 

### 3.4. Relationships between HLB Score and Leaf Morpho-Physiological Traits

Correlation and linear regression analyses were run, to identify which leaf traits are associated with HLB tolerance. Two separate correlation analyses were carried out: one for the whole population, another for the five half-sib Seedless Kishu families (Table 4). LAI was strongly correlated with HLB score for both the whole population (*r* = 0.704) as well as five half-sib seedless Kishu families (*r* = 0.771) (Table 4). Although other traits also showed significant correlations with the HLB score, the correlations were weaker than those of LAI, and r values ranged between 0.101 and 0.276. ChlF variables, PI_abs_, Fv/Fo and Fv/Fm showed a relatively stronger correlation with HLB score. Strong correlations between related variables such as LMA and leaf area as well as among Fv/Fm, Fv/Fo, and PI_abs_ were observed. HLB scores did not correlate with *Ct* value or LMA in either population (Table 4). 

As PI_abs_ showed significant association with HLB score in both populations, using linear models we tested whether the slope of the relationship varied across the quadrant direction. Interestingly, the slope of the response of PI_abs_ to HLB score was significant for all directions except the southern quadrant (Figure 4). Correlation analysis among components of PI_abs_ and HLB score (Figure 5) indicated that ψ_ET1_, γ_RC,_ and φ_Po_ correlated relatively strongly with HLB score. Additionally, ψ_ET1_and φ_Po_ components correlated strongly with PI_abs_. 

Figure 6 indicates that the first two components of PCA account for about 60% of the variation in the data. LAI and HLB score, leaf area, and leaf dry weight are positively correlated with the second component, but Fv/Fo and Fv/Fm were negatively correlated with *Ct* value and LMA in the second component (Figure 6A). On other hand, the loading plot shows that all variables are correlated positively with the first component, as noted by the positive extent in the component one dimension (Figure 6B).

A scatterplot indicates that there was a strong positive relationship between LAI and HLB score (Figure 7). To understand whether LAI can be used to predict or estimate HLB visual score, we fitted a regression line. We aimed to use LAI to predict the HLB score for HLB tolerance using simple linear regression. That is, for every 1-unit increase in LAI, the HLB score increases by 1.24 units on average. We examined the variability left over after fitting the regression line. The residuals had a constant variance, were approximately normally distributed with a mean of zero, and were independent of one another (Appendix A). The regression assumptions were met. In the summary table, the mean square error is 1.16, and R-square is 0.50. Thus, LAI explains 50% of the variation in the HLB score. The overall test results for the significance of the model are reported in the ANOVA table. The *p*-value is very small (*p* < 0.0001), so we can conclude that the model is highly significant (Appendix A).

## 4. Discussion

HLB has devastated the Florida citrus industry. The total production has declined from 240 million boxes in 2004 to 50 million boxes in the 2021–22 season (http://www.floridacitrus.org, access on 1 January 2022). Currently, in Florida, growers primarily manage HLB using production practices intended to reduce the stress of infected trees, such as insecticide applications, enhanced nutrition applications, and gibberellin [1,2]. However, no definitive solution is at hand to control HLB [1]. The optimum solution is to identify and deploy HLB-tolerant selections in production for the citrus industry. To breed HLB-tolerant cultivars, it is essential to understand the phenotypic variation and genetic effect on LB-tolerant traits. 

Previous studies have shown that a large phenotypic variation was found in the field and appeared to show field tolerance to HLB [6,7]. Several cultivars appear to have commercially useful tolerance, including some with the introgression of the mandarin genome (‘LB8-9’ Sugar Belle^®^), mandarin and citron genomes (‘Bearss’ lemon), and *Poncirus trifoliata* genome (‘US SunDragon’) [4,5]. However, genetic effect of HLB tolerance is not clear. In this study, we compared five half-sib families with Seedless Kishu as a common male parent, and found that tolerant genotypes had significant higher PI_abs_, Fv/Fm, Fv/Fo than sensitive genotypes. LB 8-9 is well-known HLB-tolerant cultivars, and its progeny was the highest among the mean of HLB score, LAI, PI_abs_, Fv/Fm, Fv/Fo and LMA than other four female parents. Genotypic variation and heritability of these parameters suggest that Fv/Fm and Fv/Fo could be used as selecting criteria of HLB-tolerant cultivars. The distribution of HLB severity visual score was in close to normal distribution. The trait varied continuously from score 1 to 9. This indicates the HLB tolerance is a polygenic trait and influenced by many gene loci, as well as environmental effect (Appendix A). Comparisons of transcriptomic, proteomic, and metabolomic profiles among tolerant and sensitive varieties revealed hundreds to thousands of genes, proteins, or metabolites associated with tolerance [18,19,20,21]. QTL mapping based on foliar symptoms and canopy damage in the field identified a few clusters of repeatable QTLs in several linkage groups of trifoliate orange that might involve hundreds of genes [13]. The effects of genetics, environment, and their interaction on phenotypic trait expression associated with HLB tolerance are complex, and knowledge of inheritance and genetic effect of HLB tolerance for selection in citrus breeding program currently is limited. In this study, we found that HLB tolerance was heritable. LAI and HLB visual score had lower heritability than other phenotypical traits due to strongly influenced by environmental effect. Fv/Fm and Fv/Fo had highest heritability which indicated that the two traits had strong genetic effect and could be potentially used for selection of HLB-tolerant selection. 

The qPCR method has been commonly used for the diagnosis of infection [14] and could be useful when trees have been infected at the early stage as measuring bacteria titer. Some studies used the quantification of bacterial titer as a measurement of HLB tolerance or HLB severity [6,7,8], but its relationship with disease severity is tenuous [22,23,24]. Although bacterial titer was tied to starch accumulation on an individual leaf basis and carbon transport on a stem segment basis, it was previously not found to correlate with symptom severity [22,23,24]. Additionally, a recent study found callose plugging (symptoms) in leaf samples that contained no *C*Las in the leaves of infected citrus trees [22]. These authors reported that most *C*Las were concentrated in the seeds and some in the root, but rarely in leaves. However, another study reported the majority of *C*Las to be located in the canopy [25]. In our study, qPCR Ct values alone provide little information to identify HLB-tolerant genotypes since all trees have been infected. We found the diagnosis of PCR-negatives accounted for 26% and 53% during December 2019 and April 2019. In addition, qPCR results only agreed between the first and second sampling for 54% of trees. This indicates apparent inconsistency between the two diagnostic qPCR results and caution use of few leaf samples to represent the whole tree. As *C*Las is not evenly distributed within a tree [26] symptoms are also heterogeneous throughout the canopy. Thus, a comprehensive rating of the whole tree including all affected areas is essential to accurately phenotype the disease response. In this study, we assume that the lack of *C*Las in the infected leaves might explain the high percentage of negative trees by qPCR, although they were infected. *C*Las reduce sieve pore callose in the leaves, and the lack of *C*las in the leaves is associated with plugged sieve pores to limit the spread of the pathogen [23]. CLas had a sharp decline in relative abundance in leaves and stems of severely affected trees [24]. In our study, there was a slightly negative correlation between qPCR Ct value and symptom severity (Table 4, Figure 2).

Reduction in canopy growth is one of the major impacts of HLB. HLB impacts phloem function, reducing the supply of sugars to growing sinks, including new shoots. Although numerous studies have measured the impacts of HLB on photosynthetic rates, the greatest impact of HLB on photosynthesis occurs in the form of reduced photosynthetic area. Diseased plants produce fewer and smaller leaves. Thus, canopy density is a measurement of the degree to which diseased plants are able to continue to form new leaf area and canopy density has positively affect fruit yield [27,28,29]. The relationship of relative yield with HLB severity can be described a negative exponential model [28]. Another study used canopy density as a proxy for disease severity within a single variety in the field and found correlations with fruit drop and other relevant disease symptoms [30]. We suggest that LAI above 4 (four times the leaf area as land area, ~ 100% light interception) can be considered as tolerant if overall the tree appears healthy. Photosynthetic activity radiation above 90% means trees had mild symptoms and high yield for Valencia and Hamlin [31]. As disease severity increases, the yield is reduced, and fruit quality degrades and yield reduction is mainly due to early abortion of fruits [29]. Given the inherent heterogeneity of symptoms, LAI considers the overall performance of the sensor and the rapid measurement, and it gives more reliable data associated with HLB tolerance than that of morphophysiological traits of a few leaves in highly variable field conditions.

Canopy density is an explicit component of HLB-disease scoring. The disease scoring used in this study includes branch dieback, leaf chlorosis, and canopy density as the factors to be assessed, thus it is not surprising that LAI would correlate well with disease severity score. However, unlike subjective disease scoring, LAI is based on the quantification of physical processes (in this case, light extinction by the canopy), and thus is less likely to be influenced by externalities such as the state of other trees surrounding those being assessed. LAI measures only canopy density, but canopy growth in height and width is sometimes captured together as canopy volume. Recently, remote sensing methods have been applied to measuring citrus tree growth in the context of HLB. Unmanned aerial vehicles in remote sensing of the growth of field trees were tested on different rootstocks under field conditions with HLB pressure [32]. A ground-based LiDAR system was used to assess canopy volume and density growth responses to different shading treatments. Such systems can enhance the power and throughput of canopy growth quantification, though ceptometry-based units, such as the one used in the present study, represent a low-cost alternative that can still phenotype hundreds of individuals per day [33]. In either case canopy density and volume measures capture an important component of disease ratings, but are unbiased and consistent between researchers, and thus represent a better approach to phenotyping. 

The physiological responses of trees induced by systemic diseases are expected to be reflected in altered functional leaf traits. While LAI represents the quantity of photosynthetic surface area, ChlF parameters can reflect photosynthetic efficiency as well as the oxidative state, or “stress,” to the photosynthetic system. Previous studies have found that photosynthesis-related genes are repressed in response to HLB [19,20,21]. ChlF parameters previously revealed a loss of photosynthetic efficiency due to HLB infection [9]. HLB affected the structure and function of photosynthetic apparatus in citrus leaves and significantly reduced the electron transport rate [10]. HLB also impaired the energetic connectivity of antennae in photosystem II (PSII) and oxygen-evolving complex, and reoxidation was inhibited. These variables are likely to be useful diagnostic markers for leaves that are strongly symptomatic, i.e., had noticeable disease symptoms [9,34,35,36,37,38,39,40]. PI_abs_ can indicate stress in plants even before the visible symptoms appear on the leaves [34]. PI_abs_ is defined as the ratio between the energy used in photosynthetic functions, and the energy that is not used in photosynthetic electron transport and dissipated as heat. In our study, PI_abs_ significantly varied across HLB severity in both whole population and five half-sib families indicating that, unlike Fv/Fm, PI_abs_ is sensitive to HLB as also reported in other plant diseases [10,32]. In the whole population, although tolerant genotypes had higher Fv/Fm, we did not find the significant difference in Fv/Fm and Fv/Fo across the HLB score [9]. The lack of significant difference for Fv/Fm and Fv/Fo in the whole population might be due to statistical noise of environment effect together with complex nature of HLB, unbalanced family size (Table 1) and genetically diverse citrus hybrid accessions (Table 2). HLB-affected trees showed reduced PI_abs_ [9]. Our Fv/Fm data were similar to values characterizing unstressed trees from other studies [12]. Components of PI, such as γ_RC_ and ψ_ET1_(probability that PSII acts as a reaction center, and efficiency of electron transfer from Q_a_ to Q_b,_ respectively) correlated strongly with HLB score and PI_abs_. Therefore, a stronger increase in PI_abs_ to HLB disease scoring was observed in those quadrants receiving less direct radiation (North, East, and West), while leaves from the south-facing quadrant were low and did not increase with greater tolerance based on the HLB score. Sensitivity of PI_abs_ to higher light intensity in citrus trees was earlier reported [9] which further supports the direction-dependent response of PI_abs_ response observed in our study. 

Although other variables assessed in this study did not exhibit the strong correlation with HLB severity in comparison with HLB visual score and LAI. However, some of these that were associated with a lesser degree shed light on certain relevant physiological aspects of photosynthesis among tolerant and sensitive types and could be further investigated, though not for purposes of time-efficient field screening and phenotyping of citrus trees for HLB sensitivity. Seedlings inoculated with *C*Las had lower LMA compared to non-inoculated plants in greenhouse conditions [11]. LMA behavior in citrus depends on the range of environmental conditions, thus it may not be surprising that the difference found under controlled conditions was not found in the present field study. The additional complexity of morphological variation among varieties further complicates the likelihood of finding a consistent effect of disease symptom severity on LMA, despite the evidence that HLB increases the LMA of specific genotypes.

It is a challenge to consistently and precisely phenotype HLB-tolerant traits in the field. Phenotyping traits with high accuracy and precision can be both costly and time-consuming. Previous studies have used different traits with different levels of categories to characterize the tolerance. The first field evaluation in 83 seed-source accessions of citrus and citrus relatives mainly from the USDA National Clonal Germplasm Repository for Citrus and Dates in Riverside, CA [8]. They used a five-rank scale for tolerance phenotype measures, whereas other researchers used six ranks [13]. Phenotyping is inconsistent among different studies because symptom severity has been subjectively assessed. For example, Sun Chu Sha and Sunki mandarins were classified in the same category as Robinson mandarin and Pineapple sweet oranges [7]. However, another study concluded that Sun Chu Sha and Sunki mandarins were more tolerant than Robinson mandarin and pineapple sweet oranges [8]. In this study, we demonstrate that tree health appearance under HLB pressure can be effectively and quantitatively evaluated using time-efficient instruments that avoid subjective human errors, while also providing information that is also biologically relevant. The most notable measurements are associated with tree canopy density measured by LAI. Additionally, photochemical PI_abs_ measures additional characteristics not captured by LAI. 

## 5. Conclusions

To effectively breed HLB-tolerant cultivars, it is critical to understand the phenotypic variation and genetic inheritance of HLB tolerance in citrus breeding programs. In this study, we found that HLB tolerance is a complex and heritable trait. With the estimation of the heritability of the HLB visual score and morphophysiological traits related to HLB tolerance, we found a significant genetic effect on HLB visual score, LAI and photosynthesis parameters, which indicates opportunities for the genetic improvement of HLB tolerance. LAI is highly correlated with HLB severity score and integrates the impact of long-term growth and necrosis as well as the photosynthetic capacity of the tree. Our study suggests that LAI and photochemical parameters could be used as a rapid and cost-effective method to evaluate HLB tolerance in citrus breeding programs.

## Figures and Tables

**Figure 1 plants-12-00042-f001:**
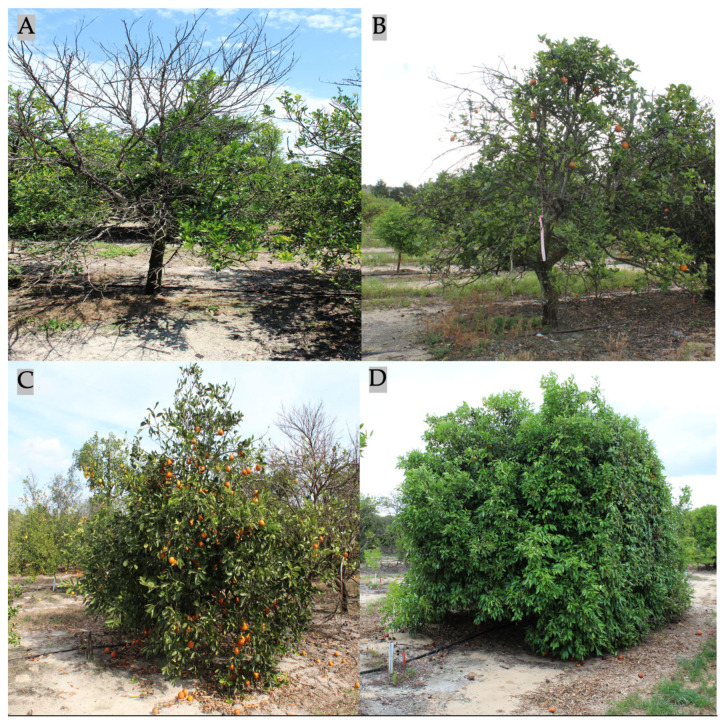
Visual healthy grading score. (**A**) Sample tree was graded score 1. (**B**) Sample tree was graded score 3. (**C**) Sample tree was graded score 5. (**D**) Sample tree was graded score 9.

**Figure 2 plants-12-00042-f002:**
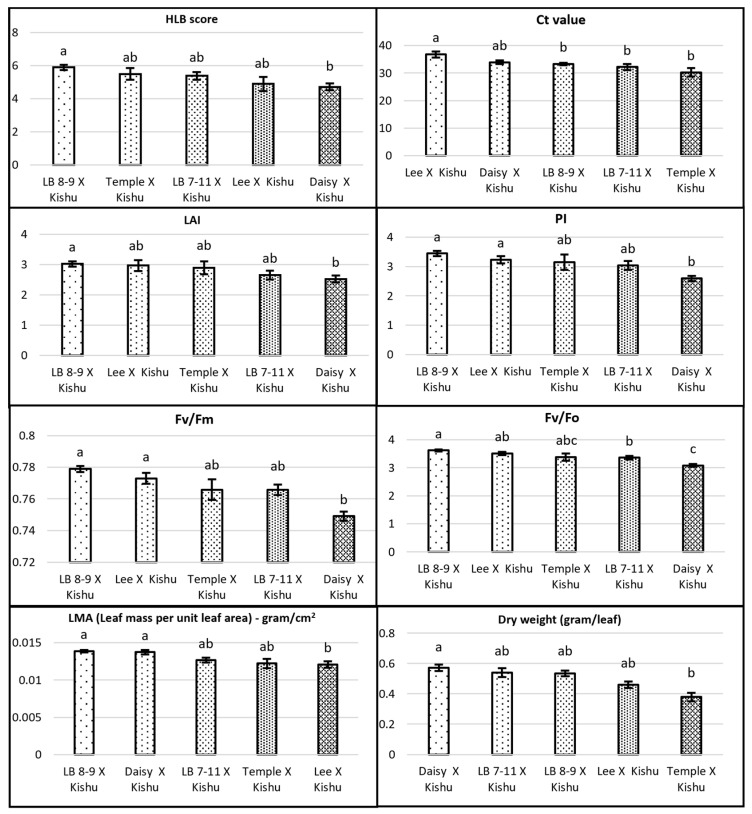
Comparison of HLB visual severity score, morphological and physiological traits among five mandarin half-sib families, Daisy × Seedless Kishu (n = 68), LB 7-11 × Seedless Kishu (n = 21), LB 8-9 × Seedless Kishu n = 100, Lee × Seedless Kishu (n = 20), Temple × Seedless Kishu (n = 12). Error bars indicate standard error. Different letters denote statistically significant differences.

**Figure 3 plants-12-00042-f003:**
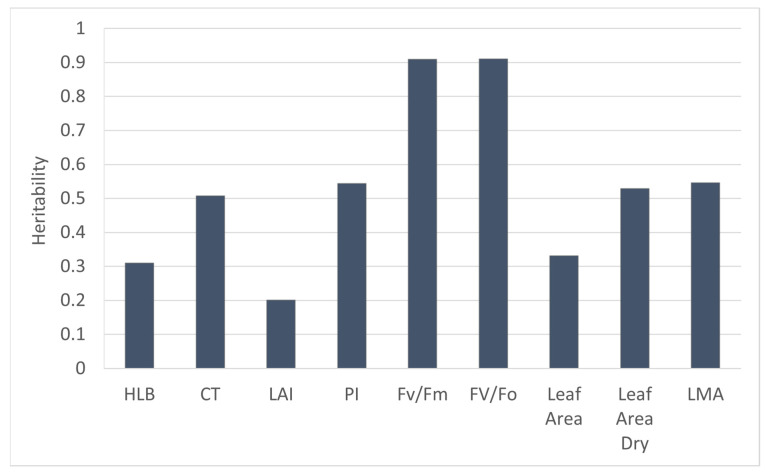
Heritability of morphological and physiological traits.

**Figure 4 plants-12-00042-f004:**
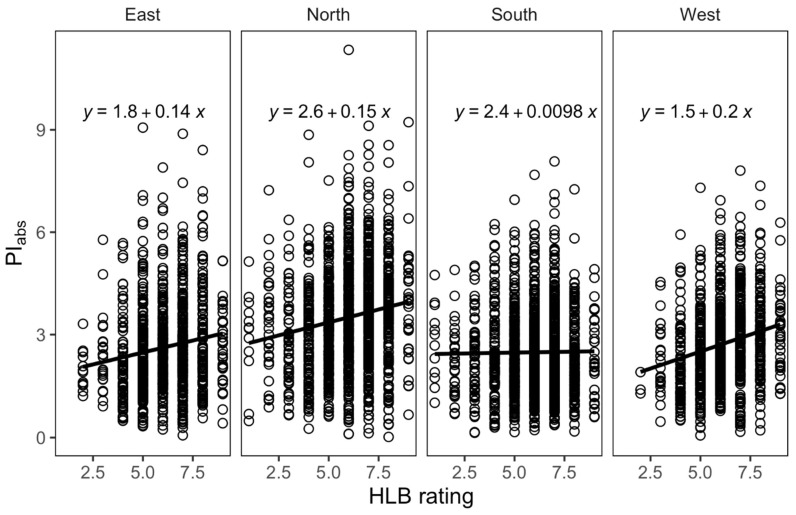
Relationship between photochemical performance index PI (PI_abs_) and HLB score across four quadrant directions (East, West, South, and North) of the tree canopy.

**Figure 5 plants-12-00042-f005:**
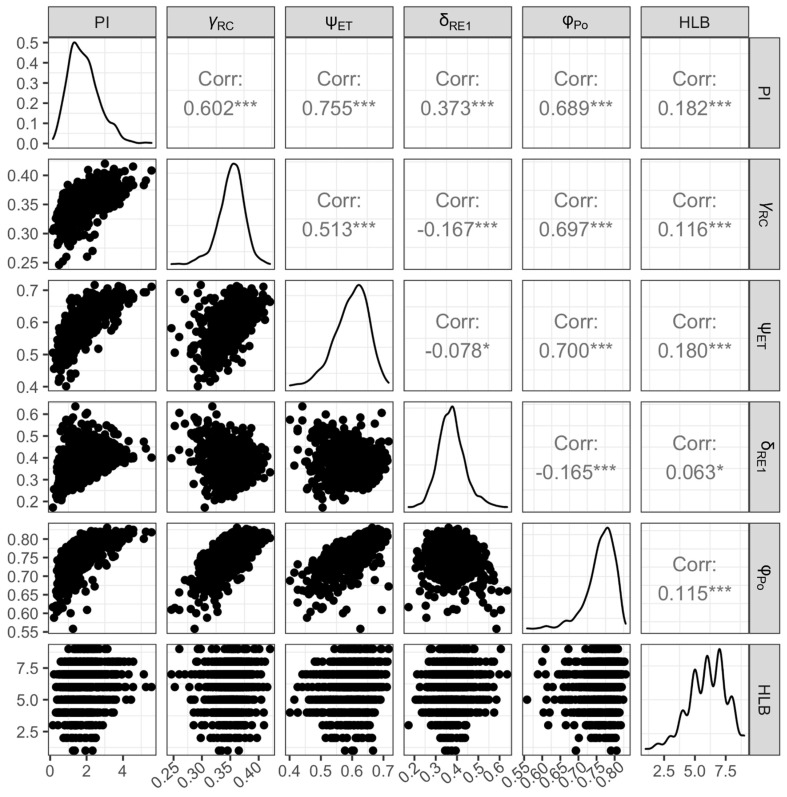
Correlation of photochemical variables, and HLB score of the studied citrus population. Stars indicate Pearson’s correlation *p* values less than 0.05. γ_RC_ probability that PSII chl functions as a reaction center, ψ_ET1_ efficiency of electron transfer from Qa to Qb, δ_RE1_ efficiency of electron transfer from Qb to PSI, φ_Po_ maximum quantum efficiency of PSII (Fv/Fm in the context of OJIP analysis). See Table 3 for more details on photochemical variables. Plots on the diagonal are integrated histograms illustrating the distribution of values along the x-axis. * denotes *p* ≤ 0.05; *** denotes *p* ≤ 0.001.

**Figure 6 plants-12-00042-f006:**
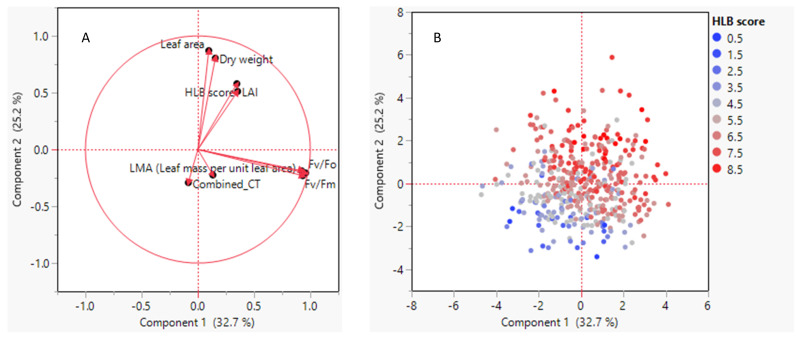
Principle component analysis. (**A**). Score pot of the first two principal components. (**B**). Loading plot showing the correlation between the original variable and the first two principal components.

**Figure 7 plants-12-00042-f007:**
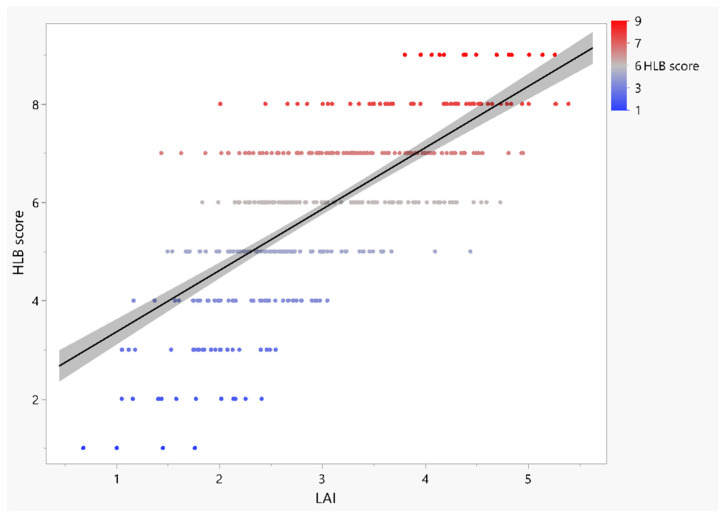
Regression analysis of HLB visual score of mandarin hybrids with leaf area indexes. Linear regression model: HLB score = 2.111378 + 1.2489657 × LAI (*p* < 0.001). R^2^ = 0.50. Significant test (*p* < 0.0001) is for intercept and LAI estimates. The grey area represents confidence intervals for the mean HLB visual score.

**Table 1 plants-12-00042-t001:** List of crosses of mandarin hybrids.

Mandarin Cross	Number of Trees	HLB Visual Score	*Ct* Value
LB 8-9 × Seedless Kishu	100	6	33
Daisy × Seedless Kishu	68	5	34
B6R9T131 × Ortanique	40	6	33
LB-5-19 × DPI Minneola	23	7	32
Lee × Seedless Kishu	20	5	37
LB 7-11 × Seedless Kishu	19	5	33
LB9-24 × Orlando	13	7	32
LB-5-19 × Ortanique	13	6	33
Temple × Seedless Kishu	13	6	31
LB8-8 × Fortune	12	6	34
King × Seedless Kishu	11	6	30
LB-8-4 × Vermilian	10	6	32
LB 8-9 × Paperrind	6	7	31
LB 8-9 × Ortanique	6	8	28
LB 2-2 × Seedless Kishu	5	5	27
LB 8-9 × Orlando Tangelo	5	6	31
B3R13T51 × Seedless Kishu	4	6	32
DPI Fortune × Minneola	4	4	28
LB9-24 × US 119	4	5	30
Lee × Fairchild	4	6	36
LB 8-8 × 9-11 USDA	3	7	33
LB 8-8 × Seedless Kishu	3	6	30
LB-7-1 × Vermilian Hong	3	7	31
Fallglo × Fairchild	2	7	29
LB 3-10 × USDA 9-11	2	7	30
LB5-19 × Sunburst	2	8	29
LB-7-1 × Bendizao	2	6	31
Robinson × Paperrind	2	7	29
Robinson × US 119	2	6	33
Temple × Orlando	2	6	30

**Table 2 plants-12-00042-t002:** List of individual mandarin hybrid accessions.

Mandarin Cross	Number of Trees	HLB Visual Score	*Ct* Value
Delta DPI 118	1	7	31
UF609	1	4	25
UF950	1	8	24
Anna (SRA)	1	7	30
Daisey	1	6	29
DPI King H4 × Marsh	1	7	31
Fallglo × Murcott	1	5	29
Fallglo/Cleo	1	5	29
Fortune × Halls	1	7	28
Fortune × Ortanique	1	3	31
Fortune × Paperrind	1	8	35
Fortune × Royal	1	5	40
Fortune × Valencia	1	5	32
Fortune/Carrizo	1	6	40
Fortune/Czo	1	8	36
Fujian Tankan	1	7	31
Kara (SRA)	1	5	31
Kawabata	1	7	33
Lakeland Limequat	1	5	28
LB 1-2 × Paperrind	1	7	29
LB-7-11	1	7	29
LB 7-5 × Imperial	1	8	26
LB 8-8/Carrizo	1	9	27
LB 8-9/Swingle	1	9	31
LB 8-9 × Big Tangelo	1	6	31
LB 8-9 × Orlando	1	9	29
LB 9-11/Carrizo	1	8	29
Lee × Murcott	1	6	37
Lee/Swingle	1	8	40
Minneola Tangelo/Swi	1	7	27
Naartje Redskin (SRA)	1	8	29
Noir/Volklmer lemon	1	7	29
Ortanique/Cleo	1	7	32
Ota/Cleo	1	6	30
Ponkan	1	6	30
Robinson × Fairchild	1	6	38
Robinson × Paperrind	1	8	31
Satsuma × Minneola	1	7	28
Satsuma × Sunburst	1	5	31
Seedless Kishu	1	6	30
Shiranui/Flying Dragon	1	3	27
Sue Linda/Carrizo	1	4	31
Tamami	1	6	32
Tankan Tangor 38752/Swi	1	8	25
Temple/Cleo	1	8	28

**Table 3 plants-12-00042-t003:** Mean of physiological and morphological traits under different group HLB visual severity scores.

HLB Score	Number of Trees	LAI	Leaf Area (cm^2^/Leaf)	Dry Weight (g)	PI_abs_	Fv/Fm	Fv/Fo	LMA	*Ct* Value
1	4	1.23 ± 0.24 d	43.6 ± 3.5 b	0.58 ± 0.07 ab	2.85 ± 0.50 abc	0.75 ± 0.03 a	3.25 ± 0.41 a	0.013 a	34.15 a
2	12	1.73 ± 0.13 d	37.95 ± 4.20 b	0.50 ± 0.05 ab	2.73 ± 0.23 abc	0.76 ± 0.01 a	3.24 ± 0.12 a	0.014 a	33.37 a
3	21	1.9 ± 0.09 d	40.99 ± 3.60 b	0.57 ± 0.05 ab	2.59 ± 0.18 bc	0.75 ± 0.01 a	3.13 ± 0.12 a	0.014 a	32.96 a
4	46	2.25 ± 0.07 d	43.72 ± 2.18 b	0.58 ± 0.03 b	2.57 ± 0.14 c	0.75 ± 0.01 a	3.14 ± 0.08 a	0.014 a	32.52 a
5	88	2.62 ± 0.06 c	45.27 ± 1.87 b	0.58 ± 0.02 b	2.72 ± 0.09 abc	0.76 ± 0.00 a	3.23 ± 0.05 a	0.013 a	32.42 a
6	104	3.1 ± 0.07 b	48.5 ± 1.80 ab	0.64 ± 0.02 ab	2.93 ± 0.09 ab	0.76 ± 0.00 a	3.31 ± 0.04 a	0.013 a	32.36 a
7	104	3.32 ± 0.07 b	50.81 ± 1.86 ab	0.67 ± 0.03 ab	3.04 ± 0.10 a	0.76 ± 0.00 a	3.37 ± 0.05 a	0.013 a	32.01 a
8	54	4.00 ± 0.10 a	57.40 ± 3.40 a	0.75 ± 0.05 a	3.06 ± 0.12 a	0.76 ± 0.00 a	3.35 ± 0.06 a	0.013 a	31.71 a
9	14	4.51 ± 0.12 a	63.73 ± 4.64 a	0.78 ± 0.05 a	3.08 ± 0.17 abc	0.76 ± 0.01 a	3.29 ± 0.11 a	0.012 a	31.38 a

Mean and standard error of mean (mean ± SEM) are used to describe the variability within the HLB visual score. Different letters denote significant differences at *p* ≤ 0.05.

**Table 4 plants-12-00042-t004:** Correlations among morphological and physiological traits in whole population and five half-sib families.

Trait	HLB Score	LAI	Leaf Area	Dry Weight	PI_abs_	Fv/Fm	Fv/Fo	*Ct* Value
Whole population								
LAI	0.704 ***							
Leaf area	0.259 ***	0.176						
Dry weight	0.228 ***	0.136	0.929 ***					
PI_abs_	0.168 ***	0.169 ***	−0.029	0.010				
Fv/Fm	0.103 *	0.124 **	−0.043	0.009	0.849 ***			
Fv/Fo	0.130 **	0.147 **	−0.019	0.032	0.907 ***	0.974 ***		
CT	−0.079	−0.132 **	−0.158 ***	−0.157 ***	−0.019	−0.023	−0.013	
LMA	−0.111	−0.128 **	−0.221 ***	0.126 **	0.078	0.101 *	0.101 *	0.006
Five seedless Kishu families								
HLB score								
LAI	0.741 ***							
Leaf area	0.104	0.055						
Dry weight	0.122	0.064	0.876					
PI_abs_	0.220 ***	0.164 ***	−0.112	−0.036				
Fv/Fm	0.228 ***	0.216 ***	−0.129	−0.056	0.893 ***			
Fv/Fo	0.226 ***	0.203 ***	−0.097	−0.021	0.917 ***	0.989 ***		
CT	0.010	−0.006	−0.231 ***	0.244 ***	0.091	0.094	0.096	
LMA	0.053	−0.024	−0.008	−0.008	0.006	0.021	0.025	−0.044

* denotes *p* ≤ 0.05; ** denotes *p* ≤ 0.01; *** denotes *p* ≤ 0.001.

## Data Availability

Not applicable.

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
