# Peer review of "Phenotypic and Genetic Variation in Morphophysiological Traits in Huanglongbing-Affected Mandarin Hybrid Populations"

_plants, 2022, doi:10.3390/plants12010042_

Round 1

Reviewer 1 Report

1- Add some information about statistical methods like correlation, PCA,  and.. into the introduction part.

2- Add a title for correlation, PCA, and,.... in the material and method section.

3- Put the references based on the journal format.

Author Response

  1. Add some information about statistical methods like correlation, PCA, . into the introduction part.

Response: The statistical methods such as ANOVA, correlation, PCA and regression analysis are described in 2. Materials and Methods, 2.6. Statistical analysis.  

  1. Add a title for correlation, PCA, and,.... in the material and method section..

Response: Under subtitle 2.6. Statistical analysis, we describes statistical methods used in this study, such as correlation and PCA.  

  1. Put the references based on the journal format.

Response: Thank Reviewer#1’s suggestion. All references are in the journal format.

Reviewer 2 Report

1. KEYWORDS - ARRANGE ALPHABETICALLY

2. FIGURE 1: HIGH-RESOLUTION PIC SHOULD BE ADDED IT PROVIDING THE BLURRY APPEARANCE

3. FIGURE 4:. Correlation of photochemical variables, and HLB score of the studied citrus population PLEASE CHECK THE VALUES IN AXIS THEY ARE INTERMINGLING WITH EACH OTHER

4. Figure 7. Regression analysis of HLB visual score of mandarin hybrids with leaf area indexes. Please add this again- BLURRY APPEARANCE

5. In this text somewhere you are using numbers for references some are using years. Follow the journal guidelines

6. References : Please follow the journal guidelines 

7. Tables : You are X or x for the hybrid crosses please use the multiplication symbol

overall manuscript written well

Author Response

Reviewer # 2

  1. KEYWORDS - ARRANGE ALPHABETICALLY.

Response: Thank Reviewer#2, Keywords are arranged alphabetically.

  1. FIGURE 1: HIGH-RESOLUTION PIC SHOULD BE ADDED IT PROVIDING THE BLURRY APPEARANCE

Response: Thank Reviewer’s suggestion. Figure 1 is in high resolution.

  1. FIGURE 4:. Correlation of photochemical variables, and HLB score of the studied citrus population PLEASE CHECK THE VALUES IN AXIS THEY ARE INTERMINGLING WITH EACH OTHER

Response: We don't understand and are happy to address anything if Reviewer can explain it.

  1. Figure 7. Regression analysis of HLB visual score of mandarin hybrids with leaf area indexes. Please add this again- BLURRY APPEARANCE

Response: Done. Figure 7 is in high resolution.

  1. In this text somewhere you are using numbers for references some are using years. Follow the journal guidelines

Response: Done. We correct the references and follow the journal guidelines.

  1. References : Please follow the journal guidelines

Response: Done. We follow the journal guideline and use the number for the references.

  1. Tables : You are X or x for the hybrid crosses please use the multiplication symbol

Response: Done.   

  1. Overall manuscript written well

Response: We want to thank Reviewer’s comments and suggestions.

Reviewer 3 Report

I have reviewed the manuscript “Phenotypic and genetic variation of morphophysiological traits in Huanglongbing affected mandarin hybrid populations”. This is a very interesting work, suitable for publication in Plants. I have the following comments:

1) Please add R2 and P-values to the Figures in this manuscript.

2) Please add a map of sampling locations to the manuscript.

3) Please add the application implications from this work.

Author Response

Reviewer # 3

  1. Please add R2 and P-values to the Figures in this manuscript.

Response: As for the regression analysis of HLB visual score and LAI in Figure 7, R2 and P value are presented in the Figure 7 legend. Other relationships are presented with correlation coefficients and P-values.

  1. Please add a map of sampling locations to the manuscript.

Response: Information of field trials are presented in Materials and Methods.

  1. Please add the application implications from this work.

Response: Application and implication of the study are discussed in the Discussion and summarized in Conclusion.